# Combined Neurological Syndrome in Electrohypersensitivity and Multiple Chemical Sensitivity: A Clinical Study of 2018 Cases

**DOI:** 10.3390/jcm12237421

**Published:** 2023-11-30

**Authors:** Dominique Belpomme, Philippe Irigaray

**Affiliations:** 1Medical Oncology Department, Paris University, 75006 Paris, France; 2European Cancer and Environment Research Institute (ECERI), 1000 Brussels, Belgium; philippei.artac@gmail.com

**Keywords:** combined sensitivity-related neurologic syndrome, electrohypersensitivity, electromagnetic field, idiopathic environmental intolerance, radiofrequency, etiopathogenic mechanism, multiple chemical sensitivity

## Abstract

From a cohort of 2018 evaluable consecutive cases issued from the European Clinical Trial Database, we describe the complete clinical symptomatic presentation of electrohypersensitivity (EHS) and multiple chemical sensitivity (MCS) and their association in the framework of a unique, sensitivity-related environmental neurologic syndrome. Eligibility criteria are those of the Atlanta consensus meeting for MCS, and those of WHO for EHS. There were 1428 EHS, 85 MCS and 505 EHS/MCS evaluable cases, so EHS was associated with MCS in 25%. Women appeared to be much more susceptible to EHS and/or to MCS than men, with no statistical significance between the EHS and MCS groups (*p* = 0.07), but the combined group revealed a more significant female sex ratio of 80.4% (*p* < 0.0001). All symptoms except emotional behavior were significantly more frequent in EHS patients than in healthy controls (*p* < 0.0001). We found no pathognomonic symptoms to establish the diagnosis of both disorders or to distinguish EHS from MCS. The three groups of patients were found to share identical symptoms, while several symptoms were found to be more significantly frequent in EHS/MCS than in EHS (*p* < 0.0001). From these data, we suggest that EHS and MCS are new brain disorders, generated via a common etiopathogenic mechanism.

## 1. Introduction

Electrohypersensitivity (EHS) and multiple chemical sensitivity (MCS) are new, acquired worldwide emerging neurologic disorders in the framework of sensitivity-related environmental illnesses. MCS was first described in 1962 by Theron G. Randolph as a disorder caused by exposure to low levels of multiple environmental chemicals [1], while EHS was identified in 1991 by William Rea and defined similarly as a pathological disorder resulting from low intensity electromagnetic field (EMF) exposure [2]. Following these pioneer works, MCS was acknowledged by the World Health Organization (WHO) during an international WHO-sponsored workshop held in 1996 in Berlin and further clinically characterized during a 1999 consensus meeting in Atlanta [3]. Likewise, EHS was acknowledged by WHO during a WHO-sponsored workshop held in 2004 in Prague [4], according to which EHS was termed idiopathic environmental intolerance to EMF, then in 2005 and 2014 in the WHO fact sheets n°296 [5] and n°193 [6], respectively, in which EHS was analyzed in connection with public health problems.

Following the initial description in 2011 by McCarthy et al. of EHS as a novel neurological syndrome [7] and our initial attempt to characterize EHS in 2015 [8], we showed that EHS can be identified as a distinct medical disorder [9], and more recently, that “it may be caused by anthropogenic electromagnetic fields (EMF) and possibly occasionally by chemicals” [10], as in MCS [3].

Because our initial observation that EHS could be associated with MCS was preliminary [8], we decided to critically review and extend this observation by investigating more patients and to clarify how and why EHS and MCS could coexist as parts of a unique, common, sensitivity-related neurological syndrome. Because, to our knowledge, there is no published exhaustive symptomatic study of both EHS and MCS in the scientific literature, we also report the complete symptomatic description of this combined syndrome in comparison with EHS and MCS alone, and on the basis of previous biological data [8,9,11] and cerebral imaging [12], we discuss a potentially common etiopathogenic mechanism involved in both disorders and in the combined syndrome. Indeed, from this study, we would like to inform physicians and scientists that they should be aware that EHS and MCS are new, frequent illnesses and that these sensitivity-related neurologic disorders are paradigmatic diseases caused by anthropogenic environments.

## 2. Material and Methods

### 2.1. Patient Accrual

Patients were not actively recruited. Accrual was due to the fact that in France, there are no medical doctors specialized in the care of EHS and MCS patients. All patients were thus spontaneously referred to one of us (DB), following their own enquiries. 

### 2.2. Inclusion Criteria

Because there is still no published study allowing a clear biological identification of MCS and EHS, we used internationally recognized clinical criteria for the inclusion of patients with either or both disorders into this study. For MCS, inclusion criteria were those internationally recognized by the 1999 Atlanta consensus meeting [3], which include that patients report being clinically intolerant to low levels of multiple environmental natural and/or man-made chemicals. We used their five defined diagnosis criteria: “(a) a chronic disorder, (b) which reproduces, (c) in response to low levels of exposure, (d) to multiple unrelated environmental chemicals, and (e) which improves when incitants are removed”. We added a sixth diagnostic clinical criterion: “symptoms occur in multiple organ systems” [13].

In our series, we have recognized two types of MCS patients: (a) those whose general chemical sensitivity was in full agreement with the above Atlanta consensus criteria, and (b) those who had more characteristic MCS clinical symptoms, such as burning and irritation of the organism’s entire airway, as had been initially reported to be due to a neurogenic inflammation [14,15,16] and more recently confirmed [17].

Our inclusion criteria for EHS patients were similar to those of the 1999 Atlanta consensus meeting for MCS, but adapted for EHS [8]. They were: (a) chronic evolution; (b) reproducibility of symptom occurrence under presumed exposure to a low-intensity level of EMF, which includes EMF emission from mobile phones, Wireless Fidelity (WiFi), powerful lines, smart meters, etc. [9]; (c) regression or disappearance of symptoms when incitants are removed; (d) absence of known pathology accounting for the observed clinical symptoms; and (e) no preexisting or coexisting pathology such as atherosclerosis, diabetes, neurodegenerative, or psychiatric diseases that would render the interpretation of clinical data difficult. Special attention was paid to exclude any cases of Alzheimer’s disease, since Alzheimer’s may be caused by EMF exposure [18,19,20,21]. Each of these EHS-related criteria were in agreement with those proposed by WHO [22]. Our criteria were not based solely on the subjective claims made by the patients, but on the clinical analysis of medical anamnesis, systematic face-to-face questioning, and on a physical examination.

### 2.3. Patient Inclusion

Patients were registered in the database we constituted and have prospectively maintained since 2009 in France, with over two thousand EHS and/or MCS self-reported cases, presently. It appears to be the most important series of such patients worldwide. This database was approved by the French Ouest VI Committee for the Protection of Persons (26 February 2018), with registration number 2017-A02706-47 and is also registered in the European Clinical Trials Database (EudraCT), with registration number 2018-001056-36. All included patients gave their informed consent for clinical research investigation and were anonymously registered. For registration, we did not use telephone interviews or internet-based questionnaire surveys, but questionnaire-driven face-to-face interviews and medical examinations, a method which minimizes patient-dependent subjective and biased or imprecise analysis.

In total, 2070 EHS and/or MCS cases were registered from 2009 to 2021, from which 2018 cases are evaluable for determination of the association of EHS with MCS and complete symptomatic analysis. The 52 non-evaluable cases, i.e., 2.5%, included 27 cases with insufficient initial clinical data and 25 which could not be correctly evaluated due to the discovery of an unrelated pathological disorder associated with EHS or MCS. Analysis was performed in 2023. Patients with EHS were compared for clinical symptoms to 100 normal controls with no MCS nor EHS and to patients with MCS or with the combined EHS/MCS syndrome.

Clinical analysis was carried out in two sequential steps. A description of the main symptoms was given in all evaluable cases, i.e., in 1428 patients with EHS, 85 patients with MCS, and 505 patients with the combined syndrome. In addition, since 2015, a more detailed analysis of symptoms was carried out on 783 patients with EHS, 51 with MCS and 307 with the combined syndrome.

### 2.4. Statistical Analysis

We used the chi-squared test for analyzing different frequency distributions, and this statistical analysis was performed using the XLSTAT software (XLSTAT 2018.1.49725; Addinsoft). The chi-squared test had a cut-off value of α = 0.05. Since chi-squared was used to perform three comparisons (EHS patients versus normal controls, EHS patients versus MCS patients and EHS patients versus EHS/MCS patients), the Bonferroni correction was applied, which sets the α cut-off of significance at 0.05/3, i.e., 0.016. 

## 3. Results

### 3.1. Frequency of MCS Associated with EHS

Our data are depicted in Table 1 and Table 2. MCS was associated with EHS in 505 cases, i.e., in 25% of the total number of EHS and MCS evaluable cases (Table 1). This result is similar to our previous 2015 report [8].

As depicted in Table 2, out the 484 combined cases evaluable for their chronological occurrence, EHS appeared first in 272 cases (56.2%), while MCS precedes EHS in 212 cases (43.8%), suggesting that in this latter case, chemicals could have been causally involved in EHS genesis, i.e., in about 10% of the total number of EHS and/or MCS cases. It is notable that many patients of the EHS group were associated with odor intolerance, albeit without reaching the standard criteria for MCS, also suggesting, nevertheless, a common hypersensitivity-associated mechanism of both disorders (see Section 4.5).

### 3.2. Demographic Data

Table 3 depicts the demographic data characterizing each EHS, MCS and EHS/MCS individualized group. A noteworthy finding is that women appear to be much more susceptible to EHS and/or to MCS than men, with no significant difference between the EHS and MCS groups, of which two thirds are female. In our series of patients, in comparison with the EHS group, there was a statistically significant pronounced female predominance of 80.4% for the combined EHS and MCS group. This suggest that females are not only prone to EHS or MCS but particularly to the combined syndrome. 

Median and mean age overall was about 49 years and did not differ statistically between the three EHS, MCS and EHS/MCS individualized groups. As indicated in Figure 1, this sample includes not only old adults but also the young and adolescents.

### 3.3. Symptomatic Presentation

We compared retrospectively the frequency of symptoms at the first clinical presentation in the EHS group of patients with that of apparently healthy people and with that of the MCS and combined EHS/MCS group of patients (Table 4). 

All clinical symptoms except emotional behavior were found to be significantly more frequent in EHS-bearing patients than in apparently healthy controls.

We observed involuntary movements of the face and of the arms simulating some pseudo-epileptic crisis, some balance disorders (specifically a Romberg sign in about 2–5% of the cases), and some paralytic ictus (the patient describes a sudden and transitory paralysis of the face or of the superior or inferior member), regardless of which of the three groups of patients is concerned. Moreover, we observed cutaneous lesions in the face, forearms or hands in 16% and 45% of the cases in EHS and EHS/MCS patients, respectively. 

Table 4 also reveals no statistically significant difference between the EHS and MCS groups for the frequency of symptoms such as headache, neck stiffness (as confirmed by cervical X-rays in all the investigated patients), skin lesions, tremors/vibrations, myalgia, trismus/neuro-muscular contraction, arthralgia, hyperacusis, photophobia, functional ocular impairment, paralytic ictus (see above), balance disorder, loss of immediate memory, confusion, fatigue, suicidal ideation, anxiety/panic crisis, emotional behavior, irritability, nausea/abdominal pain, cardiovascular abnormalities and impaired thermoregulation. 

By contrast, dysesthesia, ear heat/otalgia, tinnitus, dizziness, concentration/attention deficiency, sleep disturbance and depression tendency were statistically more frequent in EHS than in MCS patients. 

Likewise, symptoms such as skin lesions, trismus/muscular contraction, hyperacusis, ocular functional impairment, confusion, sleep disturbance, nausea/abdominal pain, chest tightness, asthma-like and ear, nose and throat (ENT) troubles were all significantly more frequent in the EHS/MCS combined syndrome than in EHS alone. This suggests that the presence of an additional chemical sensitivity component to EHS is associated with a more severe pathology. This is especially the case for skin lesions which were objectively detected in 45% of EHS/MCS patients compared to 16% of EHS patients. 

## 4. Discussion

Using the five internationally recognized Atlanta criteria for MCS, plus similar WHO-recognized criteria for EHS, in this large series of investigated patients, we have confirmed and extended our previous findings by showing that MCS is associated with EHS in about 25% of cases [8]. Thus, these two different etiopathogenic presentations may in fact be parts of a unique, common sensitivity-related syndrome.

Furthermore, we provide for the first time a complete symptomatic description of EHS, MCS and the EHS/MCS combined syndrome on the basis of a clinical analysis of a large series of 2018 EHS and/or MCS consecutive evaluable cases. There are, however, some limits to our study.

### 4.1. Study Limitations

First, we could not correlate the symptomatic clinical presentation of these disorders to a simultaneous measurement of EMFs and/or chemical exposures because of the present ubiquitous and multiform pollution of the environment. This resulted in the inclusion of patients in different groups on the basis of international clinical criteria, but not on the basis of their specific sensitivity to environmental stressors. 

Second, we used no objective biological criteria to include patients in this study, since the use of molecular biomarkers and imaging as EHS and MCS diagnosis criteria is still an open question [23].

Third, although we interviewed and physically examined all included patients, a major difficulty in our study was assessing if clinical symptoms were occurring for a low level of exposure to environmental stressors, particularly those emanating from EMF sources, so as to distinguish EHS from the idiopathic environmental intolerance (IEI) attributed to EMF exposure (IEI-EMF), which was defined during the 2004 WHO-sponsored Prague consensus meeting [4]. We thus could not prove objectively that symptom occurrence was related to low-level exposures, which we have postulated to be a characteristic distinguishing both EHS and MCS from IEI [10]; but, this was, nevertheless, clearly deduced from questioning all included patients. Measurement of all EMF sources simultaneously to symptom occurrence would be extremely difficult given the present frequent and general use of wireless technologies and the widespread diffusion of multiple chemicals in the environment.

### 4.2. Demographic Data

In this large series, young adults and adolescents were included, and a majority of them were diagnosed with EHS, whether or not associated with MCS. This may be due to their excessive use of wireless technology (mostly mobile phones, WiFi-connected computers, and other wireless devices) and to increased sensitivity to EMFs at an early age [24], as stated by WHO [5] and confirmed more recently by the American Academy of Pediatrics [25].

We have no clear explanation why the median and mean age range was 48–49 years, regardless of which EHS and/or MCS group was considered. This may be due to the exposure and/or latency period needed for these disorders to occur.

In addition in EHS and MCS, about two out three patients were females, while for the combined syndrome, this sex ratio reached 80.4%. This may reflect a genetic and/or epigenetic susceptibility of females to EMF and chemicals, particularly in combined EHS and MCS cases.

### 4.3. Symptomatic Presentation of EHS and/or MCS

In contrast to the results obtained from many self-reporting questionnaire-based studies analyzing EHS symptoms without physical examination of the patients [26,27,28,29,30,31,32,33,34,35], we did not find that all symptoms were subjective. Our data are thus in contrast with these studies and with the official statement by the WHO [5], which was neither based on studies involving medical face-to-face clinical interviews, nor on neurological and general physical examinations of the patients.

Surprisingly, many of these studies focused on the symptomatic risk in EMF-exposed people from the general population, with only a few studies having focused specifically on symptoms in EHS self-reported patients. To our knowledge, no study described the complete picture of EHS and/or MCS patients. Moreover, all the general population-related studies were based on telephone surveys or mailed or web-based questionnaires, not on face-to-face questioning and physical examination. We recently summarized all known, original published studies reporting symptoms which may occur in EHS patients, as well as in healthy people submitted to EMF exposure; their selection procedures were different among studies, and the EMF sources are often not well characterized [10].

These studies erroneously conclude that symptoms in EHS patients or in healthy people exposed to EMF are purely subjective, differ from one to another individual, and are not related to EMF exposure [31,33,36,37]. Unfortunately these studies did not look for an association between EHS and MCS, which may conflate their symptomatic dependency on EMF exposure, i.e., that symptoms could have occurred due to a low concentration of environmental chemicals and not necessarily to EMF exposure alone.

Indeed, since the seminal identification of EHS by William Rea [2], clinical abnormalities have been described in mobile phone users by Bruce Hocking [38], while this author and Roderick Westerman documented neurological changes in C-fiber nerves induced by mobile phone exposure [39]. Furthermore, in a double-blind provocation study performed with a single EHS case, Mc Carty et al. showed that neurological EMF-associated clinical symptoms constitute a novel neurological syndrome [7]. This provocation study employed extremely low frequencies of EMF-exposure (60 Hz, 300 V/m electric field, continuous or 10 Hz on/off pulsations) specifically designed to minimize unintentional sensory reactions, with symptoms causally attributed primarily to off–on/on–off exposure transitions, rather than to uninterrupted EMF exposure.

### 4.4. Toward a Medical Assessment of EHS and/or MCS as Acquired Environmental Sensitivity-Related Somatic Neurological Disorders

In the present scientific literature, the clinical symptoms reported by EHS patients (unlike those reported by MCS patients) are not considered as true medical symptoms, but simply claimed to be “self-reported symptoms”. This is contrary to medical practice carried out since Hippocrates. This misclassification may be due to the use of telephone- or web-based survey analysis to inform investigators on symptoms put forward by the patients, rather than by questioning and examining patients. Hence, scientists have not used a valuable medical descriptive tool to identify and diagnose EHS. As can be soundly deduced from any face-to-face medical questioning and by physical examination, there is, a priori, no medical reason to dismiss a patient’s live conditions and to assume that patients invent or mislabel when they attribute their symptoms to anthropogenic EMF and/or chemical exposure.

In this study, contrary to other reports [36,37], we have shown that not all clinical symptoms are subjective, and that most of them are reproducible from one patient to another, with no fundamental difference for clinical presentations between the three groups.

In addition, we have found that EHS is associated with MCS in 25% of cases, and we have previously argued that EHS could be causally associated with anthropogenic EMF exposure, and possibly with environmental chemicals, as in MCS [9]. We have shown also that both EHS and MCS can be characterized by identical biomarkers detected in the peripheral blood and urine of patients [8,11]. Therefore, contrary to previous reports [40,41], we strongly suggest that EHS and MCS are objective somatic disorders which cannot be hypothesized to originate from non-EMF-related psychologic or psychiatric causes or from vague, undefined functional impairments [10,23].

In this study, we have shown there are presently no pathognomonic clinical symptoms which allow a clear distinction between EHS and MCS. Although most symptoms (but not all) are subjective and considered not specific, the overall clinical picture resulting from their prevalence and from their association strongly supports that EHS and MCS, and the combined syndrome, can be identified as new, typical neurologic disorders, regardless of their causal origin. As shown in Table 4, the six most characteristic common symptoms co-occurring in EHS and MCS cases are headache, dysesthesia, tinnitus, dizziness, ear heat/otalgia and cognitive deficiency.

More recently, an interesting study by Frederic Greco focused on the prevalence of migraine in EHS patients [42]. However, we should add to migraine-associated symptoms symptoms not classically involved in migraine disease, such as tinnitus, balance disorder and cognitive deficiency. Likewise, myalgia and muscular spasm (not to be confused with fibromyalgia), transitory cardiovascular symptoms, skin lesions (not to be confused with allergic erythema), chronic fatigue (not to be confused with the chronic fatigue syndrome), and depressive tendency all might be added to the EHS and/or MCS symptomatic core reported here.

### 4.5. EHS and/or MCS as New Brain Disorders

On the basis of the present clinical symptom analysis, and elsewhere of cerebral imaging [12] and brain neurotransmitter concentration measurement in urine [9], we have previously provided strong arguments for a predominant pathological role of the brain in EHS and MCS. The fact that EHS can be frequently associated with MCS strongly suggests that the mechanism of EHS genesis in the brain may involve the olfactory-limbic system, as it has been suggested to be involved in MCS [14]. This does not exclude, however, effects of EMF and/or chemicals on other parts of the organism.

Moreover, the neurologic symptomatology associated with EHS and/or MCS (see Table 4) suggests impairment of CNS synaptic plasticity, as synapses play a key role in the transmission of neuronal electrochemical signals [43]. It has been shown that the N-methyl-D-aspartate (NMDA) receptor, one of the ionotropic glutamate receptors widespread expressed in the CNS, may play a crucial role in synaptic transmission, and that its over-expression is closely associated with impaired synaptic plasticity occurring in different neuro-pathophysiological processes [44,45] and may be involved in MCS [46]. It has been shown experimentally that the NMDA receptor signaling pathway could be over-expressed in the rat hippocampus following microwave exposure [47]. This finding reinforces our hypothesis of a role of the olfactory limbic system, not only in MCS, but also in EHS and the combined syndrome.

## Figures and Tables

**Figure 1 jcm-12-07421-f001:**
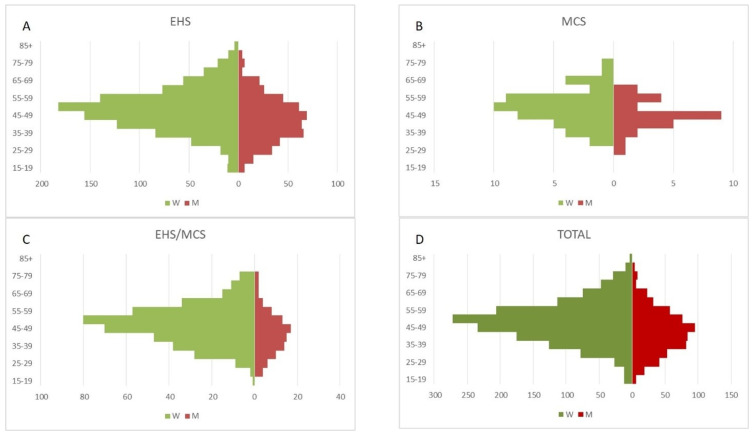
Relative number of patients in different age categories: the EHS group (**A**) compared to the MCS group (**B**) and EHS/MCS combined group (**C**), and the total 2018 investigated cases (**D**).

**Table 1 jcm-12-07421-t001:** Frequency of the Association of MCS with EHS.

	2015 Analysis	Present Analysis
	Evaluable Cases *n* = 727	Evaluable Cases *n* = 2018
EHS	521/727 (71.7%)	1428/2018 (70.8%)
MCS	52/727 (7.1%) *	85/2018 (4.2%) *
EHS/MCS	154/727 (21.2%) * 154/675 (22.8%) **	505/2018 (25%) * 505/1933 (26.1%) **

* Ratio determined from the all patients studied, i.e., including patients of the MCS group. ** Ratio determined from the EHS patients studied, i.e., without patients of the MCS group.

**Table 2 jcm-12-07421-t002:** Ratio and percentage of EHS patients who later suffered from MCS and vice versa.

Total number of evaluable EHS/MCS combined cases *	484
EHS patients that latter suffered of MCS	272/484 (56.2%)
MCS patients who latter suffered of EHS	212/484 (43.8%)

* In 21 cases, the occurrence of EHS and MCS was simultaneous. They were omitted from the analysis.

**Table 3 jcm-12-07421-t003:** Demographic data based on 2018 serially individualized cases of EHS and/or MCS.

Demographic Data	EHS	MCS	*p* *	EHS/MCS	*p* **
Number of cases (%)	1428 (70.8%)	85 (4.2%)	-	505 (25%)	-
Age (mean ± SD)	48.86 ± 12.74	49.15 ± 9.71	0.84	48.57 ± 11.38	0.67
Age (median (range))	49 (16–85)	48 (29–77)	-	49 (19–76)	-
Sex ratio (women/men)	972/456 (68%)	50/35 (58.8%)	0.07	406/99 (80.4%)	<0.0001

Chi-squared test. * Comparing the MCS group of patients to that of EHS. ** Comparing the EHS/MCS group of patients to that of EHS.

**Table 4 jcm-12-07421-t004:** Frequency of clinical symptoms in EHS-bearing patients in comparison with that in seemingly healthy individuals and that in MCS and EHS/MCS patients *.

Clinical Symptoms	EHS Ratio (%)	Normal Controls (%) *n* = 100	*p* **	MCS Ratio (%)	*p* ***	EHS/MCS Ratio (%)	*p* ****
Headache	1285/1428 (90%)	13	**<0.0001**	76/85 (89%)	0.86	454/505 (90%)	0.95
Neck stiffness *	251/783 (32%)	0	**<0.0001**	16/51 (31%)	0.92	101/307 (33%)	0.79

Dysesthesia	1200/1428 (84%)	0	**<0.0001**	60/85 (71%)	**0.001**	374/505 (74%)	**<0.0001**
Skin lesions	228/1428 (16%)	0	**<0.0001**	15/85 (18%)	0.68	227/505 (45%)	**<0.0001**
Tremors/vibrations *	157/783 (20%)	0	**<0.0001**	5/51 (10%)	0.07	61/307 (20%)	0.95

Myalgia *	360/783 (46%)	6	**<0.0001**	22/51 (43%)	0.69	154/307 (50%)	0.21
Trismus/muscular contraction *	62/783 (8%)	0	**<0.0001**	1/51 (2%)	0.12	49/307 (16%)	**0.0004**
Arthralgia *	250/783 (32%)	19	**0.008**	14/51 (27%)	0.51	83/307 (27%)	0.11

Ear heat/otalgia *	493/783 (63%)	0	**<0.0001**	22/51 (43%)	**0.004**	166/307 (54%)	**0.0013**
Tinnitus	914/1428 (64%)	5	**<0.0001**	34/85 (40%)	**<0.0001**	313/505 (62%)	0.42
Hyperacusis	500/1428 (35%)	6	**<0.0001**	26/85 (31%)	0.41	278/505 (55%)	**<0.0001**

Photophobia *	260/783 (33%)	0	**<0.0001**	14/51 (27.5%)	0.92	90/307	0.41 (29.4%)
Ocular troubles	472/1428 (33%)	0	**<0.0001**	38/85 (45%)	0.03	222/505 (44%)	**<0.0001**

Paralytic ictus *	78/783 (10%)	0	**<0.0001**	7/51 (14%)	0.39	46/307 (15%)	0.02
Dizziness	985/1428 (69%)	0	**<0.0001**	43/85 (51%)	**0.0004**	303/505 (60%)	**0.0002**
Balance disorder	485/1428 (34%)	**0**	**<0.0001**	30/85 (35%)	0.80	116/505 (23%)	**<0.0001**

Concentration/attention deficiency	1114/1428 (78%)	0	**<0.0001**	55/85 (65%)	**0.004**	424/505 (84%)	**0.002**
Loss of immediate memory	1085/1428 (76%)	6	**<0.0001**	57/85 (67%)	0.06	394/505 (78%)	0.35
Confusion *	47/783 (6%)	0	**<0.0001**	2/51 (4%)	0.54	46/307 (15%)	**<0.0001**

Sleep disturbance	1071/1428 (75%)	6	**<0.0001**	43/85 (51%)	**<0.0001**	409/505 (81%)	**0.006**
Fatigue	1200/1428 (84%)	10	**<0.0001**	41/51 (80%)	0.49	444/505 (88%)	0.04

Depression tendency	814/1428 (57%)	0	**<0.0001**	26/85 (30%)	**<0.0001**	237/505 (47%)	**<0.0001**
Suicidal ideation	229/1428 (16%)	0	**<0.0001**	9/85 (11%)	0.18	91/505 (18%)	0.30
Anxiety/panic	372/1428 (26%)	0	**<0.0001**	31/85 (36%)	0.03	152/505 (30%)	0.08
Emotional behavior	186/1428 (13%)	11	0.56	13/85 (15%)	0.55	75/505 (15%)	0.30
Irritability	328/1428 (23%)	6	**<0.0001**	15/85 (18%)	0.25	126/505 (25%)	0.37

Nausea/abdominal pain *	141/783 (18%)	0	**<0.0001**	8/51 (16%)	0.67	101/307 (33%)	**<0.0001**
Cardiovascular abnormalities	657/1428 (46%)	0	**<0.0001**	36/85 (42%)	0.51	253/505 (50%)	0.11
Chest tightness *	94/783 (12%)	2	**<0.0001**	29/51 (56%)	**<0.0001**	172/307 (56%)	**<0.0001**
Asthma-like crisis *	47/783 (6%)	0	**<0.0001**	22/51 (43%)	**<0.0001**	132/307 (43%)	**<0.0001**
ENT (ear, nose, and throat) troubles *	94/783 (12%)	4	**<0.0001**	44/51 (86%)	**<0.0001**	92/307 (30%)	**<0.0001**
Impaired thermoregulation	186/1428 (13%)	0	0.02	5/85 (6%)	0.05	25/505 (5%)	**<0.0001**

*n*: number of evaluable cases. *p*: probability that difference is due to random variation. * Since 2015, more sophisticated symptoms have been analyzed in EHS patients and compared retrospectively with symptoms obtained from a series of 100 apparently normal subjects used as controls. These symptoms were also compared to those occurring in MCS and EHS/MCS patients (see Material and Methods). Percentages of patients with symptoms compared with the chi-square independence test. ** *p*-value between EHS patients and normal controls. *** *p*-value between EHS patients and MCS patients. **** *p*-value between EHS patients and EHS/MCS patients.

## Data Availability

Data are contained within the article.

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
