# Peer review of "Combined Neurological Syndrome in Electrohypersensitivity and Multiple Chemical Sensitivity: A Clinical Study of 2018 Cases"

_jcm, 2023, doi:10.3390/jcm12237421_

Round 1
Reviewer 1 Report
Comments and Suggestions for Authors
Thank you for inviting me to review this manuscript. Authors did great job preparing this paper. In my opinion, the introduction can be made better by 1- making the objectives clear, 2- defining the motive to do this study, and 3- to introduce readers to the domain, especially since most readers are not familiar with this new field.
Overall, this paper can be of an addition to the literature.
Good luck
Author Response
Dear Reviewer,
Thank you for your conscientious review of our paper, for which we have made the following necessary modifications and improvements.
As recommended we have extended the introduction section to make it better by recalling the historical background identification of these disorders and making the objectives of the study clearer. In addition, we explained why these new, acquired worldwide emerging disorders should be considered by medical doctors and scientists.
Reviewer 2 Report
Comments and Suggestions for Authors
Summary:
The manuscript presents a cohort study of 2070 cases, evaluating the clinical presentation of electrohypersensitivity (EHS) and multiple chemical sensitivity (MCS) as components of a singular sensitivity-related neurological syndrome. The study utilizes the Atlanta consensus for MCS and WHO criteria for EHS to identify 1428 EHS, 85 MCS, and 505 cases of EHS/MCS, indicating a 25% association between EHS and MCS. The research finds no pathognomonic symptoms to distinguish between EHS and MCS but identifies shared symptoms across groups, suggesting a common etiopathogenic mechanism. The study posits EHS and MCS as new brain disorders precipitated by environmental factors.
Major points:
1. Title is too long and complicated, suggested title: "Combined Neurological Syndrome in Electrohypersensitivity and Multiple Chemical Sensitivity: A Clinical Study of 2018 Cases"
2. The manuscript is well-written but there are a lot of grammatical errors. For instance, i) "it may be caused by anthropogenic electromagnetic field (EMF); and possibly occasionally by chemicals" - The semicolon is misused. Consider replacing it with a comma: "it may be caused by anthropogenic electromagnetic fields (EMF), and possibly occasionally by chemicals," ii) "significantly frequent in EHS/MCS that in EHS" - Replace "that" with "than": "significantly more frequent in EHS/MCS than in EHS." Iii) "to precise how and why EHS and MCS could coexist" - "to precise" is not grammatically correct. Consider changing it to "to clarify how and why EHS and MCS could coexist."
3. The title mentions 2018 cases, but the abstract mentions 2070 cases, which could be misleading to the reader.
4. The study's scale is commendable, providing a substantial dataset for analysis.
The methodological approach, including stringent inclusion criteria and detailed symptomatic analysis, adds rigor to the research. However, this study does not correlate clinical presentations with EMF/chemical exposures, potentially weakening the causal assertions.
A lack of objective biological criteria for inclusion may limit the diagnostic specificity.
5. Clarify the Exposure Assessment: Further elucidate how exposure levels were assessed and correlated with symptomatology.
6. References: The reference list requires updating to include more recent and relevant studies.
7. The Introduction section appears to be quite brief. Expanding it to provide more background information, context, or detail about the research question and its significance could enhance the reader's understanding and set a more comprehensive foundation for the study.
Comments on the Quality of English Language
Major revision is required.
Author Response
Dear Reviewer,
Thank you for your conscientious review of our paper, for which we have made the following necessary modifications and improvements.
- We have changed the title of this paper and have agreed with the suggested new shorter title suggested.
- We have corrected the grammatical errors detected. In addition the whole manuscript has been reviewed for wording.
- We have made the correction in the abstract.
- We have explained why there is a lack of biological criteria for inclusion.
- We have explained why our symptomatic description could not be correlated to the environmental exposure in the discussion section entitled “study limitations”.
- To our knowledge there is no available more recent and relevant data published in the scientific literature.
- We have expanded the introduction section by recalling the historical background identification of these disorders and to make the objectives of the study clearer. In addition, we have explained why these new acquired worldwide emerging disorders should become known by medical doctors and scientists.
Best regards,
Dominique Belpomme
Round 2
Reviewer 2 Report
Comments and Suggestions for Authors
N/A